# The GET pathway serves to activate Atg32-mediated mitophagy by ER targeting of the Ppg1-Far complex

Mashun Onishi, Mitsutaka Kubota, Lan Duan, Yuan Tian, Koji Okamoto

**Mitophagy removes defective or superfluous mitochondria via selective autophagy. In yeast, the pro-mitophagic protein Atg32 localizes to the mitochondrial surface and interacts with the scaffold protein Atg11 to promote degradation of mitochondria. Although Atg32-Atg11 interactions are thought to be stabilized by Atg32 phosphorylation, how this posttranslational modification is regulated remains obscure. Here, we show that cells lacking the guided entry of the tail-anchored protein (GET) pathway exhibit reduced Atg32 phosphorylation and Atg32-Atg11 interactions, which can be rescued by additional loss of the ER-resident Ppg1-Far complex, a multi-subunit phosphatase negatively acting in mitophagy. In GET-deficient cells, Ppg1-Far is predominantly localized to mitochondria. An artificial ER anchoring of Ppg1-Far in GET-deficient cells significantly ameliorates defects in Atg32-Atg11 interactions and mitophagy. Moreover, disruption of GET and Msp1, an AAA-ATPase that extracts non-mitochondrial proteins localized to the mitochondrial surface, elicits synthetic defects in mitophagy. Collectively, we propose that the GET pathway mediates ER targeting of Ppg1-Far, thereby preventing dysregulated suppression of mitophagy activation.**

## Introduction

Mitochondria-specific autophagy, termed mitophagy, is one of the membrane trafficking pathways conserved from yeast to humans. In this process, mitochondria are sequestered by flattened double-membrane structures called isolation membranes and transported to the lysosome (in mammals) or the vacuole (in yeast), a lytic compartment, for degradation (Palikaras et al, 2018; Onishi et al, 2021; Onishi & Okamoto, 2021). In the budding yeast *Saccharomyces cerevisiae*, the outer mitochondrial membrane (OMM)–anchored protein Atg32 is phosphorylated in a manner dependent on casein kinase 2 under mitophagy-inducing conditions (Kanki et al, 2009, 2013; Okamoto et al, 2009; Aoki et al, 2011; Kondo-Okamoto et al, 2012). This post-translational modification increases the affinity of

Atg32 for Atg11 (Aoki et al, 2011; Kondo-Okamoto et al, 2012; Kanki et al, 2013), a scaffold protein for assembly of core autophagy–related (Atg) proteins required for formation of autophagosomes selectively encapsulating specific cargoes (Kim et al, 2001; Shintani et al, 2002; Suzuki et al, 2002). Conversely, Atg32 dephosphorylation is mediated by Ppg1, a PP2A-like phosphatase (Furukawa et al, 2018). Ppg1 interacts with the Far complex, co-operatively acting in suppression of Atg32 phosphorylation, Atg32-Atg11 interactions, and mitophagy (Furukawa et al, 2018). Far9 and Far10, two components of the Far complex, contain tail-anchored (TA) domains and are required for localization of Ppg1-Far to the ER and mitochondria (Pracheil & Liu, 2013; Innokentev et al, 2020). OMM anchoring of Ppg1-Far via Far9 is required for Atg32 dephosphorylation (Innokentev et al, 2020).

Appropriate targeting of membrane proteins to correct subcellular destinations is critical to maintain functional compartments within cells (Barlowe & Miller, 2013). TA proteins, which harbor a single transmembrane (TM) domain at the very C-terminus, are post-translationally inserted into the membranes of mitochondria, peroxisomes, and ER, acting in a myriad of cellular processes such as vesicular trafficking, protein import, and organelle dynamics (Barlowe & Miller, 2013). In budding yeast, multiple TA proteins are targeted to the ER via the guided entry of the TA protein (GET) pathway (Denic, 2012; Denic et al, 2013; Farkas & Bohnsack, 2021). Before insertion into the ER membrane, the TM domains of TA proteins are shielded by the cytosolic ATPase Get3 (Bozkurt et al, 2009; Mateja et al, 2009, 2015; Suloway et al, 2009; Yamagata et al, 2010). Then, the Get3-TA protein complexes are recruited to the ER membrane–embedded Get1/2 insertase complex (Schuldiner et al, 2008; Stefer et al, 2011; Wang et al, 2011, 2014; McDowell et al, 2020). Successful interactions between Get1/2 and Get3 drive detachment of TA proteins from Get3, enabling their insertion into the ER membrane by the Get1/2 complex. Upon disruption of the GET pathway, several TA proteins are not properly localized to the ER, but, instead, targeted to mitochondria (Schuldiner et al, 2008; Jonikas et al, 2009). These ER-resident TA proteins on the OMM are removed by Msp1, a mitochondrial surface–anchored AAA-ATPase that extracts inappropriately targeted non-mitochondrial TA proteins and thus maintains mitochondrial

Laboratory of Mitochondrial Dynamics, Graduate School of Frontier Biosciences, Osaka University, Suita, Japan

Correspondence: okamoto.koji.fbs@osaka-u.ac.jp
Mashun Onishi's present address is Max Planck Institute for Biology of Ageing, Cologne, Germany

membrane integrity (Zhang et al, 2011; Chen et al, 2014; Okreglak & Walter, 2014; Wohlever et al, 2017; Wang et al, 2020; Matsumoto et al, 2022).

Our previous findings reveal a previously unappreciated role for Get1/2 in promoting mitophagy during prolonged respiratory growth (Onishi et al, 2018). In contrast to severely impaired mitophagy, other selective and bulk autophagy pathways are only slightly affected, indicating that the common core autophagy machinery itself is rarely altered in the absence of Get1/2 (Onishi et al, 2018). Although it is likely that the Get1/2 complex serves a specialized function in mitophagy, how this ER-resident TA protein insertase acts in degradation of mitochondria remains uncertain. In this study, we demonstrate that Atg32 phosphorylation and Atg32-Atg11 interactions are compromised in cells lacking Get components. Notably, perturbation of Ppg1-mediated Atg32 dephosphorylation mostly recovers Atg32-Atg11 interactions and mitophagy in *get1/2*-null cells. Moreover, Ppg1-Far is localized to the ER in a manner dependent on the GET pathway, and loss of the Get components leads to predominant targeting of this phosphatase complex to mitochondria. Artificial ER localization of Far9 in the absence of Get1/2 significantly restores Atg32-Atg11 interactions and mitophagy. In addition, disruption of Msp1 extractase activity in GET-deficient cells causes an exacerbation in mitophagy defects. Taken together, our data suggest that the GET pathway serves to promote appropriate targeting of the Ppg1-Far complex to the ER, thereby contributing to Atg32 activation at the initial stage of mitophagy.

# Results

## Atg32 phosphorylation and Atg32-Atg11 interactions are reduced in cells lacking GET components

In yeast, mitophagy initiation consists of three main steps, expression, mitochondrial localization, and phosphorylation of Atg32. Based on our previous results that Get components are not critical for Atg32 expression and mitochondrial localization (Onishi et al, 2018), we sought to test whether loss of Get components affects Atg32 phosphorylation under mitophagy-inducing conditions. When grown in media containing non-fermentable carbon sources such as glycerol, cells contain mitochondria that are respiratory active. In those cells reached to the post-log phase under respiratory conditions, a substantial fraction of mitochondria is transported to the vacuole and degraded via mitophagy (Okamoto et al, 2009). During this process, Atg32 is phosphorylated in the early phase of respiratory growth (Kondo-Okamoto et al, 2012; Kubota & Okamoto, 2021). Phosphorylated Atg32 molecules appeared as multiple upper bands (Fig 1A) that were diminished by treatment with a protein phosphatase (Fig 1C). In contrast, these mobility shifts seemed to be reduced in *get1*-, *get2*-, or *get3*-null cells (60–70% of WT cells), indicating that Get components are important for efficient phosphorylation of Atg32 (Fig 1A and B).

As Atg32 phosphorylation is thought to be a key regulatory step for stabilizing Atg32-Atg11 interactions (Aoki et al, 2011; Kondo-Okamoto et al, 2012), we next investigated whether loss of Get components impinges this protein–protein interaction for mitophagy. To address this issue, we applied the NanoLuc Binary

Technology (NanoBiT; Promega) system, a luminescence-based assay using split luciferase consisting of LgBiT and SmBiT, to quantitative monitoring of Atg32-Atg11 interactions in live cells. When yeast cells expressing chromosomally integrated LgBiT-tagged Atg32 and SmBiT-tagged Atg11 were grown under respiratory conditions, the Atg32-Atg11 interaction brings the LgBiT and SmBiT subunits into close proximity, resulting in reversible reconstitution of an active luciferase that generates a luminescent signal in the presence of its substrate furimazine (Dixon et al, 2016) (Fig S1A). This system, which efficiently drives mitophagy (80% compared with WT cells) without overexpression, enables us to measure the resulting luminescent signals by a microplate reader and relatively quantify Atg32-Atg11 interactions in vivo. Our NanoBiT system detected lower luminescent signals in cells lacking Get1, Get2, or Get3 (threefold to fivefold reduction compared with WT cells) under respiratory conditions (Fig 1D). Reduction in Atg32-Atg11 interactions did not seem to be mainly caused by a decrease in Atg32 and Atg11 expression levels (Fig S1B and C). Thus, these results indicate that Get components are required for promoting Atg32-Atg11 interactions.

## Perturbation of the Ppg1 phosphatase restores Atg32-Atg11 interactions and mitophagy in get1/2-null cells

It is conceivable that a decrease in Atg32 phosphorylation causes suppression of Atg32-Atg11 interactions in cells lacking Get components (Fig 1A–D). Thus, we hypothesized that augmentation of Atg32 phosphorylation could rescue the impaired protein–protein interactions for mitophagy in GET-deficient cells. To test this possibility, we attempted to genetically increase Atg32 phosphorylation by loss of Ppg1, a protein phosphatase acting in dephosphorylation of Atg32 and suppression of Atg32-Atg11 interactions (Furukawa et al, 2018). Accordingly, we performed NanoBiT assays and found that consistent with the previous report (Furukawa et al, 2018), loss of Ppg1 increased Atg32-Atg11 interactions (twofold to threefold compared with WT cells) (Fig 2A). Remarkably, in *get1/2 ppg1*-double-null cells, Atg32 interacted with Atg11 at near WT levels, supporting the idea that reduced Atg32 phosphorylation in cells lacking Get1/2 is the primary cause of a defect in Atg32-Atg11 interactions (Fig 2A).

Next, we performed mitophagy assays using the mitochondrial matrix–localized DHFR-mCherry (mito-DHFR-mCherry) probe (Calvelli et al, 2020). When mitochondria are transported to the vacuole, DHFR-mCherry is processed by vacuolar proteases to generate free mCherry, enabling semi-quantitative detection of mitochondrial degradation. We confirmed that loss of Ppg1 accelerated mitophagy (137% compared with WT cells) (Fig 2B and C). Strikingly, *get1 ppg1*- and *get2 ppg1*-double-null cells exhibited mitophagy at near WT levels (112% and 89%, respectively, compared with WT cells) (Fig 2B and C). Moreover, the expression of a *PPG1 H111N* gene encoding a catalytically inactive phosphatase restored Atg32-Atg11 interactions and mitophagy in *get1/2*-null cells (Fig S1D–F). Together, these results suggest that perturbation of Ppg1 increased the affinity of Atg32 for Atg11 in *get1/2*-null cells, thereby recovering mitophagy.

To exclude the possibility that restoration of mitophagy in *get1 ppg1*- and *get2 ppg1*-double-null cells is caused indirectly by pleiotropic

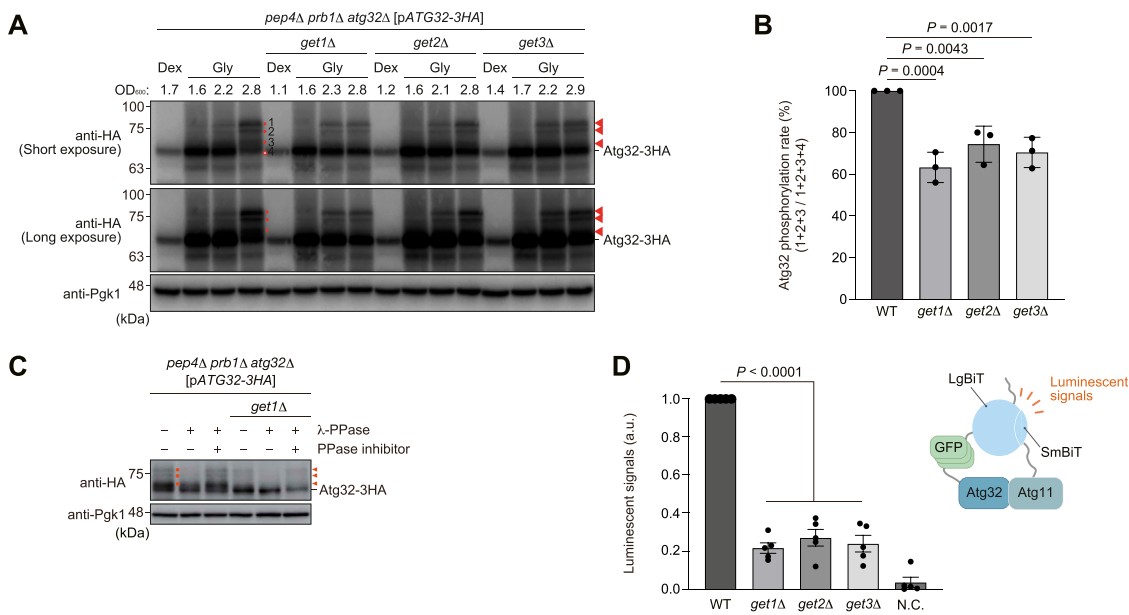

**Figure 1.  Atg32 phosphorylation and Atg32-Atg11 interactions are reduced in cells lacking Get components.**
**(A)** WT, *get1Δ*, *get2Δ*, and *get3Δ* cells containing a plasmid encoding Atg32-3HA (p*ATG32-3HA*) pregrown in fermentable dextrose medium (Dex) were cultured in non-fermentable glycerol medium (Gly), collected at the indicated $OD_{600}$ points, and subjected to Western blotting. All strains are *pep4 prb1 atg32*-triple-null derivatives defective in vacuolar degradation of Atg32-3HA via mitophagy. Atg32 is phosphorylated at the early stages of respiratory growth, and phosphorylated Atg32 molecules are detected as multiple upper protein bands (Nos. 1–3). Orange arrowheads and dots indicate putative phosphorylated Atg32. Pgk1 was monitored as a loading control.
**(A, B)** Intensities of phosphorylated Atg32 (Nos. 1–3) in (A) were normalized to total Atg32 protein intensities (Nos. 1–4). The signal values in WT cells at the 2.8 $OD_{600}$ points were set to 100%. Data represent the averages of all experiments (*n* = 3 independent cultures, means ± s.e.m.). **(C)** *pep4Δ prb1Δ atg32Δ* and *pep4Δ prb1Δ atg32Δ get1Δ* cells containing a plasmid encoding Atg32-3HA were grown in glycerol medium, collected at the $OD_{600}$ = 2.5 point, and subjected to alkaline lysis and TCA precipitation. The pellet was resuspended in a reaction buffer, treated with or without lambda protein phosphatase (*λ*-PPase) in the presence or absence of PPase inhibitor. Orange arrowheads and dots indicate putative phosphorylated Atg32. **(D)** WT, *get1Δ*, *get2Δ*, and *get3Δ* cells expressing Atg32 internally tagged with 3×GFP plus Large BiT and Atg11 C-terminally tagged with Small BiT or cells expressing Atg32 and Atg11 (negative control, N.C.) were grown in glycerol medium, collected at the $OD_{600}$ = 1.4 point, incubated with substrates, and subjected to the NanoBiT-based bioluminescence assays. Luminescent signals in WT cells were set to 1. Data represent the averages of all experiments (*n* = 5 independent cultures, mean ± s.e.m.). a.u., arbitrary unit. **(B, D)** Data were analyzed by one-way ANOVA with Dunnett's multiple comparison test (B, D). Source data are available for this figure.

alterations in Ppg1 substrate(s), we examined Atg32 variants lacking the amino acid residues 151–200 that are required for the Ppg1-Far complex to interact with Atg32 (Furukawa et al, 2018; Innokentev et al, 2020). When this truncation was introduced into the NanoBiT system, the Atg32 mutant (Δ151–200) interacted with Atg11 14–18-fold more strongly than the full-length protein in the presence of Get1, and at near WT levels even in the absence of Get1 (Fig 2D). Consistent with these results, mitophagy in *get1/2*-null cells was mostly restored by the expression of the Atg32 mutant (Δ151–200) (Fig 2E and F). We also confirmed that the expression of the Atg32 mutant (Δ151–200) does not significantly change in *get1/2*-null cells (Fig S1G), suggesting that these phenotypes are not mainly caused by aberrant Atg32 levels. Collectively, these data support the idea that Ppg1-Far–mediated suppression of Atg32-Atg11 interactions and mitophagy is exacerbated in the absence of Get1/2.

## The Far complex predominantly targets to mitochondria in GET-deficient cells

How could Ppg1 abrogate mitophagy in cells lacking Get1/2? It has been demonstrated that ER-resident TA proteins localize to

mitochondria in *get1/2*-null cells (Schuldiner et al, 2008; Jonikas et al, 2009). In addition, Ppg1 interacts with the Far complex that acts in pheromone-induced cell cycle arrest and the TORC2 signaling pathway (Kemp & Sprague, 2003; Pracheil et al, 2012; Furukawa et al, 2018). Moreover, the Far complex contains the TA proteins Far9 and Far10 and is anchored to the ER membrane in a manner dependent on their TA domains (Pracheil & Liu, 2013). Based on these findings, we hypothesized that disruption of the GET pathway may lead to exclusive targeting of the Ppg1-Far complex to the surface of mitochondria, thereby oversuppressing mitophagy. To test this idea, Far8, a component of the Far complex, was functionally tagged with three copies of GFP, expressed from the chromosomal *FAR8* locus without overexpression, and observed using fluorescence microscopy. We found that Far8-3×GFP mostly colocalized with mCherry-tagged Sec63, an ER-anchored Hsp40/DnaJ family protein (Feldheim et al, 1992) that exhibited peripheral and perinuclear patterns, in WT cells under respiratory conditions (Fig 3A and B). In contrast, Far8-3×GFP predominantly localized to mitochondria in *get1*-null cells (93% of cells lacking Get1 and 9% of WT cells) (Fig 3C and D). We also confirmed that loss of Get2 or Get3 greatly increased mitochondria-targeted Far8-3×GFP (96% and 92% of *get2*- and *get3*-null cells, respectively) (Fig S2A).

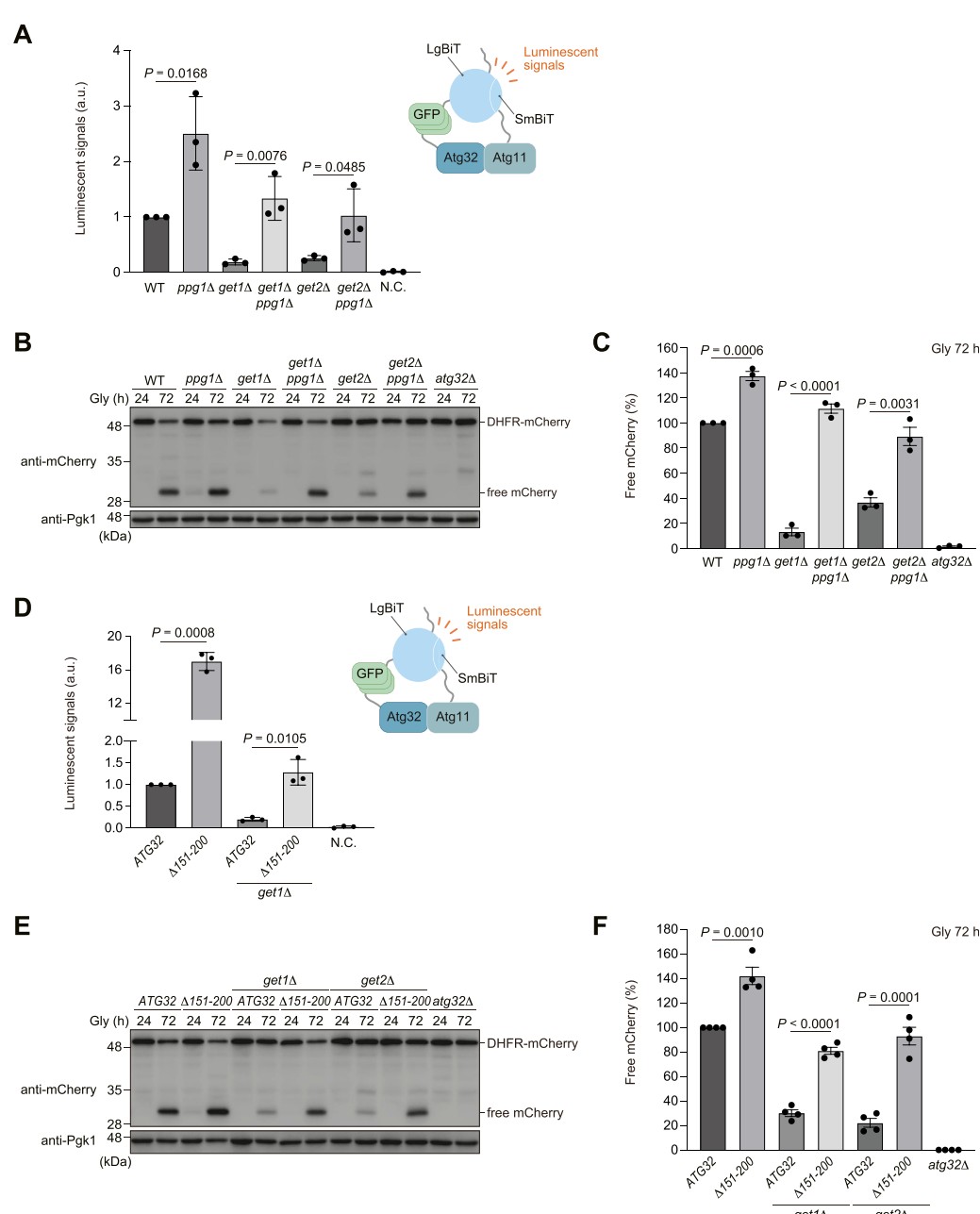

**Figure 2. Perturbation of the Ppg1 phosphatase restores Atg32-Atg11 interactions and mitophagy in *get1/2*-null cells.**
**(A)** WT, *ppg1Δ*, *get1Δ*, *get2Δ*, *get1Δ ppg1Δ*, and *get2Δ ppg1Δ* cells expressing Atg32-3HA-3×GFP-3FLAG-LgBiT and Atg11-HA-SmBiT, or cells expressing Atg32 and Atg11 (negative control, N.C.) were grown in glycerol medium (Gly), collected at the $OD_{600}$ = 1.4 point, incubated with substrates, and subjected to the NanoBiT-based bioluminescence assays. Luminescent signals in WT cells were set to 1. Data represent the averages of all experiments (*n* = 3 independent cultures, means ± s.e.m.). a.u., arbitrary unit. **(B)** Mitochondria-targeted DHFR-mCherry–expressing (mito-DHFR-mCherry) WT, *ppg1Δ*, *get1Δ*, *get2Δ*, *get1Δ ppg1Δ*, *get2Δ ppg1Δ*, and *atg32Δ* cells were grown for the indicated time points in glycerol medium (Gly) and subjected to Western blotting. Generation of free mCherry indicates transport of the marker to the vacuole. **(B, C)** Amounts of free mCherry in cells under respiratory conditions in (B) were quantified in three experiments. The signal intensity value of free mCherry in WT cells at the 72-h time point was set to 100%. Data represent the averages of all experiments (*n* = 3 independent cultures, means ± s.e.m.). **(D)** WT and *get1Δ* cells expressing Atg11-HA-SmBiT and Atg32-3HA-3×GFP-3FLAG-LgBiT or (Δ151–200)-3HA-3×GFP-3FLAG-LgBiT, or cells expressing Atg32 and Atg11 (negative control, N.C.) were grown in glycerol medium (Gly), collected at the $OD_{600}$ = 1.4 point, incubated with substrates, and subjected to the NanoBiT-based bioluminescence assays. Luminescent signals in WT cells were set to 1. Data represent the averages of all experiments (*n* = 3 independent cultures, means ± s.e.m.). a.u., arbitrary unit. **(E)** WT, *get1Δ*, and *get2Δ* cells expressing chromosomally integrated *ATG32* WT or *ATG32* (*Δ151–200*), and *atg32Δ* cells were grown in glycerol medium (Gly) and subjected to Western blotting. All strains were derivatives expressing mito-DHFR-mCherry. **(E, F)** Amounts of free mCherry in cells under respiratory conditions in (E) were quantified in four experiments. The signal intensity value of free mCherry in WT cells at the 72-h time point was set to 100%. Data represent the averages of all experiments (*n* = 4 independent cultures, means ± s.e.m.). **(A, C, D, F)** Data were analyzed by a two-tailed *t* test (A, C, D, F).
Source data are available for this figure.

To clarify whether the insertase activity of Get1/2 is required for Far8-3×GFP localization to the ER, we generated yeast strains expressing an inactive Get1 or Get2 variant with point mutations in their conserved cytosolic domain (Get1NRm: N72A, R73A; Get2RERRm: R14E, E15R, R16E, R17E) (Wang et al, 2011), and confirmed that these mutants are expressed at near WT levels (Fig S2B and C). Using these strains, we

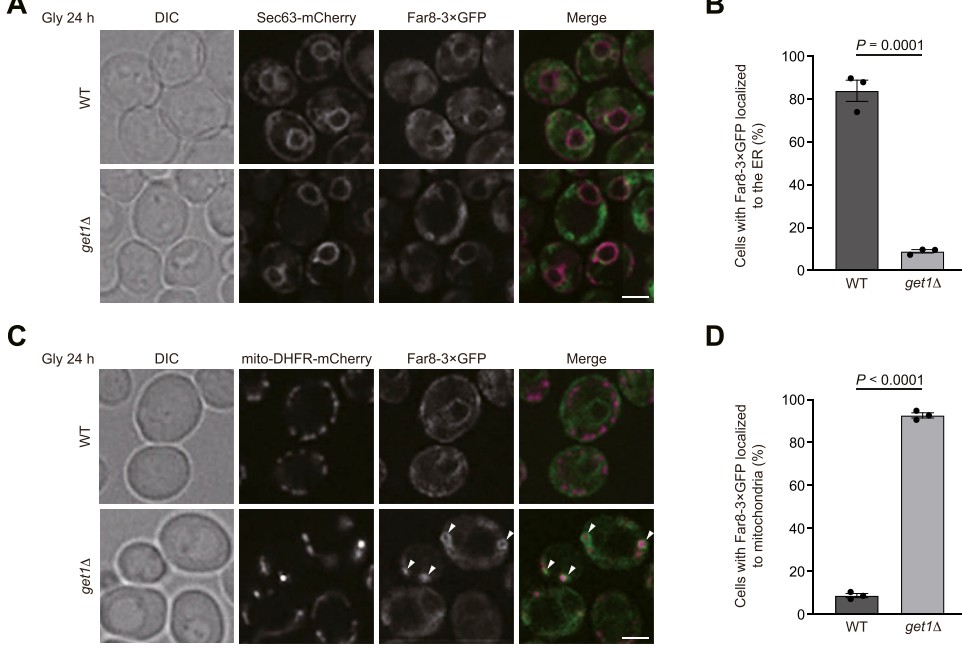

**Figure 3. Far8 predominantly targets to mitochondria in cells lacking Get1.**
**(A)** Representative images of WT and *get1Δ* cells expressing Sec63-mCherry and Far8-3×GFP grown for 24 h in glycerol medium (Gly) and observed by structured illumination microscopy. Single-plane images are shown. Scale bar, 2 μm. DIC, differential interference contrast. **(A, B)** Cells analyzed in (A) were quantified in three experiments. Data represent the averages of all experiments (*n* = 3 independent cultures, means ± s.e.m.). **(C)** Representative images of WT and *get1Δ* cells expressing mito-DHFR-mCherry and Far8-3×GFP grown for 24 h in glycerol medium (Gly) and observed by structured illumination microscopy. Single-plane images are shown. Arrowheads indicate Far8-3×GFP localized to mitochondria. Scale bar, 2 μm. **(C, D)** Cells analyzed in (C) were quantified in three experiments. Data represent the averages of all experiments (*n* = 3 independent cultures, means ± s.e.m.). **(B, D)** Data were analyzed by a two-tailed *t* test (B, D). Source data are available for this figure.

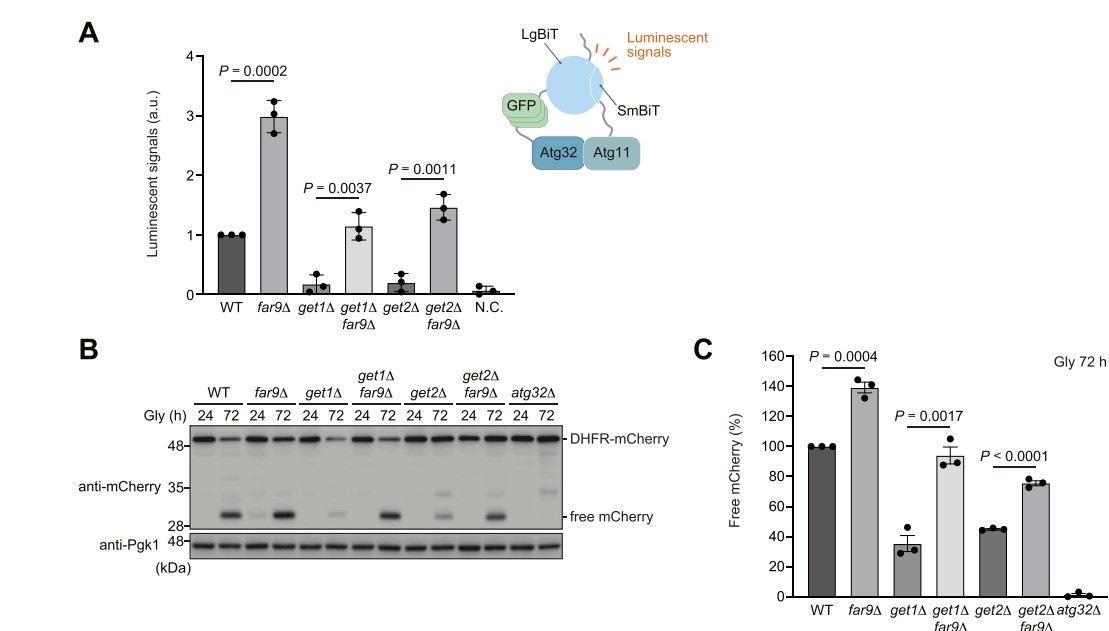

**Figure 4. Loss of Far9 restores Atg32-Atg11 interactions and mitophagy in *get1/2*-null cells.**
**(A)** WT, *far9Δ*, *get1Δ*, *get2Δ*, *get1Δ far9Δ*, and *get2Δ far9Δ* cells expressing Atg32-3HA-3×GFP-3FLAG-LgBiT and Atg11-HA-SmBiT, or cells expressing Atg32 and Atg11 (negative control, N.C.) were grown in glycerol medium (Gly), collected at the OD$_{600}$ = 1.4 point, incubated with substrates, and subjected to the NanoBiT-based bioluminescence assays. Luminescent signals in WT cells were set to 1.0. Data represent the averages of all experiments (*n* = 3 independent cultures, means ± s.e.m.). a.u., arbitrary unit. **(B)** WT, *far9Δ*, *get1Δ*, *get2Δ*, *get1Δ far9Δ*, *get2Δ far9Δ*, and *atg32Δ* cells expressing mito-DHFR-mCherry were grown in glycerol medium (Gly), collected at the indicated time points, and subjected to Western blotting. **(B, C)** Amounts of free mCherry in cells under respiratory conditions in (B) were quantified in three experiments. The signal intensity value of free mCherry in WT cells at the 72-h time point was set to 100%. Data represent the averages of all experiments (*n* = 3 independent cultures, means ± s.e.m.). **(A, C)** Data were analyzed by a two-tailed *t* test (A, C). Source data are available for this figure.

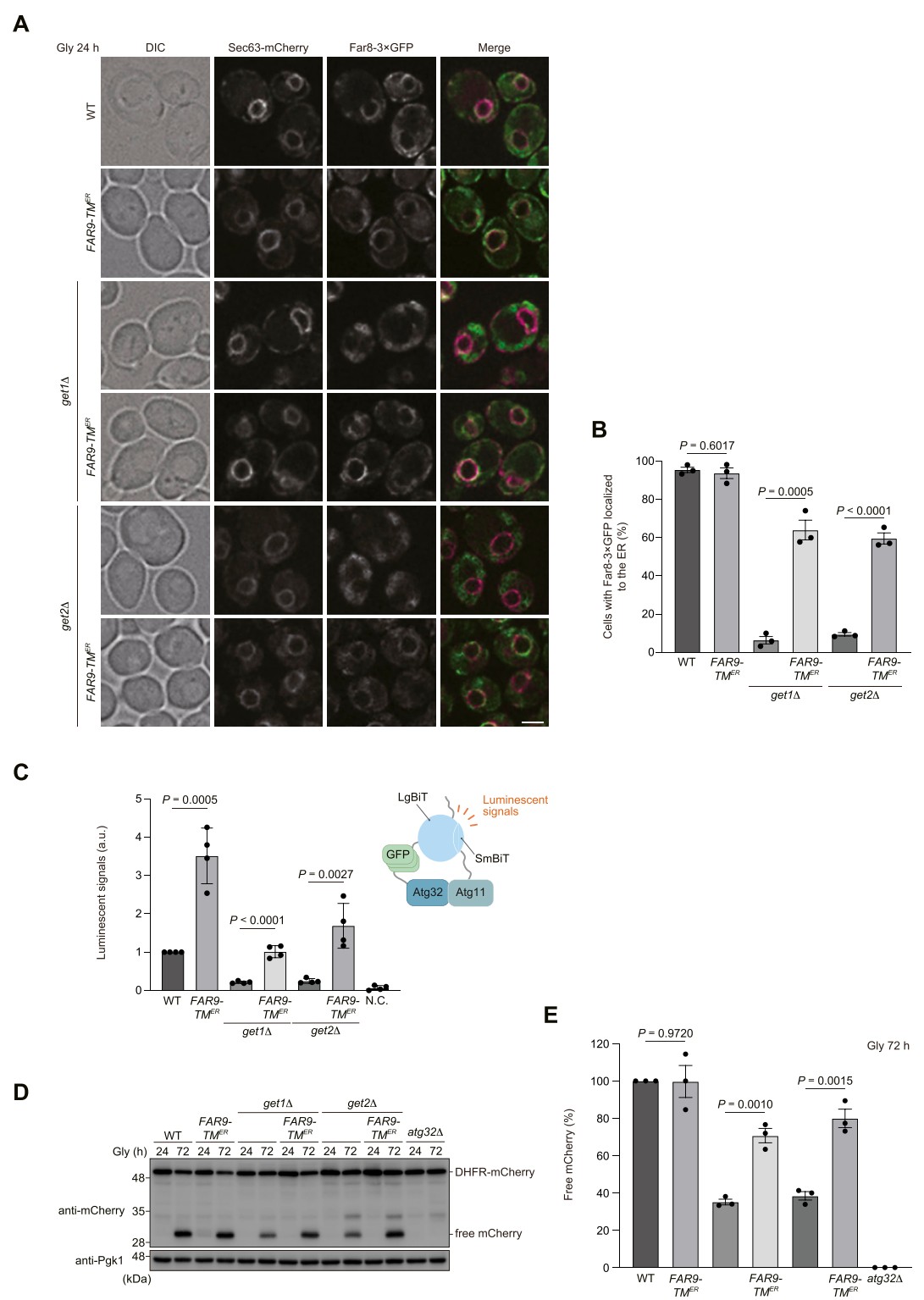

**Figure 5. Forced ER anchoring of Far9 ameliorates mitophagy deficiencies in cells lacking Get1/2.**
**(A)** Representative images of WT, *get1Δ*, and *get2Δ* cells expressing the endogenous Far9 or a variant whose TA domain was replaced with the Sec12 TM domain (*FAR9-TM$^{ER}$*) grown for 24 h in glycerol medium (Gly) and observed by structured illumination microscopy. Single-plane images are shown. All strains were derivatives expressing Sec63-mCherry and Far8-3×GFP. Scale bar, 2 μm. **(A, B)** Cells analyzed in (A) were quantified in three experiments. Data represent the averages of all experiments (*n* = 3 independent cultures, means ± s.e.m.). **(C, D)** Derivatives of cells analyzed in (D) expressing Atg32-3HA-3×GFP-3FLAG-LgBiT and Atg11-HA-SmBiT, or cells expressing Atg32 and Atg11 (negative control, N.C.) were grown in glycerol medium (Gly), collected at the OD$_{600}$ = 1.4 point, incubated with substrates, and subjected to the NanoBiT-based

found that the expression of these insertase-inactive mutants significantly disturbed ER localization of Far8-3×GFP (95% and 98% of cells expressing Get1NRm and Get2RERRm, respectively) (Fig S2D), further underscoring a primary role for the GET pathway in ER targeting of the Ppg1-Far complex. In cells expressing these mutants, mitophagy was moderately reduced (Fig S2E and F), indicating that the Get1/2 insertase activity is required for efficient mitophagy.

Because targeting of ER-resident TA proteins to the mitochondrial surface requires their TA domains (Farkas & Bohnsack, 2021), we assumed that loss of Far9 or Far10 could diminish mitochondrial localization of the Far complex in cells lacking Get1/2. In line with this idea, we found that Far8-3×GFP was hardly localized to mitochondria, but instead mostly dispersed throughout the cytoplasm (probably excluded from the vacuolar lumen) in *far9/10*-null, *far9 get1*- and *far10 get1*-double-null cells (Fig S3A and B), indicating that these TA proteins are indispensable for targeting of the Far complex to the ER (and perhaps in addition to mitochondria) in WT cells, or mitochondria in GET-deficient cells.

It has recently been reported that a fraction of the Far complex is localized to mitochondria even in WT cells under fermentable conditions (Innokentev et al, 2020). Although we barely found mitochondrial localization of Far8-3×GFP under non-fermentable conditions (Fig 3A and C), it remained possible that a small fraction of the Far complex is localized to mitochondria and degraded in a mitophagy-dependent manner. To clarify this issue, we performed GFP-processing assays. Similar to mito-DHFR-mCherry, Far8-3×GFP can be transported to the vacuole and processed to generate free GFP via ER-phagy and mitophagy. Under respiratory conditions, the generation of free GFP was reduced by 50% in cells without mitophagy (*atg32*-null) or ER-phagy (*atg39 atg40*-double-null) (Mochida et al, 2015) and 25% in cells without both events (*atg32 atg39 atg40*-triple-null) compared with WT cells (Fig S3C and D). These results support the notion that a small fraction of the Ppg1-Far complex escapes the GET pathway and localizes to the surface of mitochondria.

### Loss of the Far9/10 TA proteins rescues mitophagic deficiencies in cells lacking Get1/2

Our observations that mitochondrial localization of the Far complex in *get1*-null cells was diminished by loss of Far9 or Far10 (Fig S3A and B) led us to examine Atg32-Atg11 interactions and mitophagy in the absence of these TA proteins. Similar to the results obtained from *ppg1*-null cells (Fig 2A), Atg32 interacted with Atg11 twofold to threefold more strongly in cells lacking Far9 than WT cells (Fig 4A). In addition, consistent with the previous findings (Furukawa et al, 2018), mitophagy under respiratory conditions was increased in *far9*-null cells (139% compared with WT cells) (Fig 4B and C). Strikingly, Atg32-Atg11 interactions and mitophagy were restored at near WT levels in *get1 far9*- and *get2 far9*-double-null cells (Fig 4A–C).

Next, we investigated cells lacking Far10 and found only a slight and no increase in Atg32-Atg11 interactions and mitophagy (1.2-fold and 98%, respectively, compared with WT cells) (Fig S4A–C). Notably, *get1 far10*- and *get2 far10*-double-null cells exhibited a partial recovery in Atg32-Atg11 interactions (0.7- and 0.4-fold, respectively, compared with WT cells) (Fig S4A) and a substantial restoration in mitophagy (96% and 69%, respectively, compared with WT cells) (Fig S4B and C). Collectively, these data suggest that the Ppg1-Far complex is anchored to the mitochondrial surface via Far9/10 and acts in suppression of Atg32-Atg11 interactions and mitophagy.

### Artificial ER anchoring of the Far complex increases Atg32-Atg11 interactions and mitophagy in get1/2-null cells

Based on our findings that loss of Get1/2 leads to excess mitochondrial localization of the Ppg1-Far complex (Figs 3A–D and S2A), we asked whether Get1/2-independent ER localization of the Ppg1-Far complex ameliorates mitophagy deficiencies in cells lacking Get1/2. To this end, the TA domain of Far9 was replaced with the TM domain (TM$^{ER}$) of Sec12, a single-pass ER membrane protein consisting of an N- and C-terminal domains facing the cytosol and ER lumen, respectively (d'Enfert et al, 1991). We confirmed that the expression of Far9-TM$^{ER}$ does not cause significant alterations in ER shape and Far8-3×GFP localizations (Fig 5A and B). As expected, Far8-3×GFP in cells expressing Far9-TM$^{ER}$ was localized to the ER even in *get1/2*-null cells (Figs 5A and B and S5A and B). In addition, the expression of Far9-TM$^{ER}$ in cells lacking Get1/2 restored Atg32-Atg11 interactions at near WT levels (Fig 5C). Moreover, mitophagy was increased in *get1*- and *get2*-null cells (70% and 80%, respectively, compared with WT cells) (Fig 5D and E), suggesting that ER retention of the Ppg1-Far complex is critical for efficient mitophagy.

### Artificial mitochondrial anchoring of the Far complex partially reduces mitophagy

As excess accumulation of the Far complex on the mitochondrial surface by loss of Get1/2 seems to perturb mitophagy, we sought to test whether artificial targeting of the Far complex to mitochondria may suppress mitophagy without disrupting Get1/2 functions. The TA domains of Far9 and Far10 were replaced with those derived from Gem1 (Frederick et al, 2004), an OMM protein (Far9/Far10-TA$^{MITO}$). We confirmed that Far8-3×GFP almost exclusively localizes to mitochondria in cells expressing Far9/Far10-TA$^{MITO}$ (Fig 6A and B). In these cells, Atg32-Atg11 interactions and mitophagy under respiratory conditions were partially reduced (0.5-fold and 70% compared with WT cells, respectively) (Fig 6C–E). Importantly, this reduction was mostly abrogated in cells lacking Ppg1 or expressing

bioluminescence assays. Luminescent signals in WT cells were set to 1. Data represent the averages of all experiments (*n* = 4 independent cultures, means ± s.e.m.). a.u., arbitrary unit. **(D)** WT, *get1Δ*, *get2Δ*, and *atg32Δ* cells expressing mito-DHFR-mCherry and WT *FAR9* or *FAR9-TM$^{ER}$* were grown in glycerol medium (Gly), collected at the indicated time points, and subjected to Western blotting. **(D, E)** Amounts of free mCherry in cells under respiratory conditions for 72 h in (D) were quantified in three experiments. The signal intensity value of free mCherry in WT cells at the 72-h time point was set to 100%. Data represent the averages of all experiments (*n* = 3 independent cultures, means ± s.e.m.). **(B, C, E)** Data were analyzed by a two-tailed *t* test (B, C, E).
Source data are available for this figure.

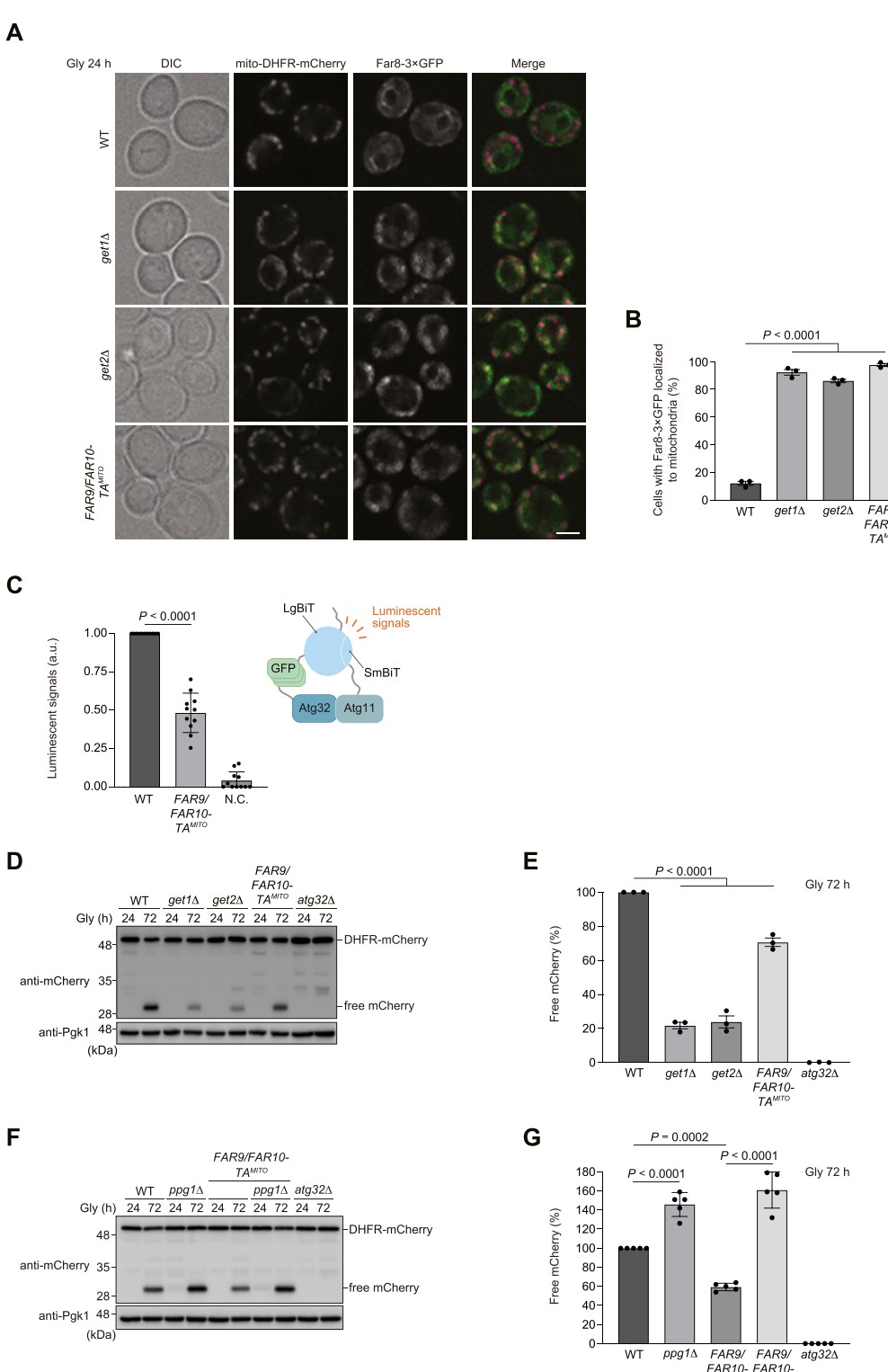

**Figure 6. Forced mitochondrial anchoring of Far9/Far10 partially reduces Atg32-Atg11 interactions and mitophagy.**

**(A)** Representative images of WT, *get1Δ*, and *get2Δ* cells expressing the endogenous Far9/Far10 or a variant whose TA domains were replaced with the Gem1 TM domain (*FAR9/FAR10-TM^MITO^*) grown for 24 h in glycerol medium (Gly) and observed by structured illumination microscopy. Single-plane images are shown. All strains were derivatives expressing mito-DHFR-mCherry and Far8-3×GFP. Scale bar, 2 μm. **(A, B)** Cells analyzed in (A) were quantified in three experiments. Data represent the averages of all experiments ($n$ = 3 independent cultures, means ± s.e.m.). **(C)** Endogenous Far9/Far10- or Far9/Far10-TA^MITO^–expressing cells harboring Atg32-3HA-3×GFP-3FLAG-LgBiT and Atg11-HA-SmBiT, or cells expressing Atg32 and Atg11 (negative control, N.C.) were grown in glycerol medium (Gly), collected at the OD$_{600}$ = 1.4 point, incubated with substrates, and subjected to the NanoBiT-based bioluminescence assays. Luminescent signals in WT cells were set to 1. Data represent the averages of all experiments ($n$ = 11 independent cultures, mean ± s.e.m.). a.u., arbitrary unit. **(D)** WT, *get1Δ*, *get2Δ*, and *atg32Δ* cells expressing mito-DHFR-mCherry and the endogenous Far9/Far10 or Far9/Far10-TA^MITO^ were grown in glycerol medium (Gly), collected at the indicated time points, and subjected to Western blotting. **(D, E)** Amounts of free mCherry in cells under respiratory conditions for 72 h in (D) were quantified in three experiments. The signal intensity value of free mCherry in WT cells at the 72-h time point was set to 100%. Data represent the averages of all experiments ($n$ = 3 independent cultures, means ± s.e.m.). **(F)** WT, *ppg1Δ*, and *atg32Δ* cells expressing mito-DHFR-mCherry and the endogenous Far9/Far10 or Far9/Far10-TA^MITO^ were grown in glycerol medium (Gly), collected at the indicated time points, and subjected to Western blotting. **(F, G)** Amounts of free mCherry in cells under respiratory conditions for 72 h in (F) were quantified in five experiments. The signal intensity value of free mCherry in WT cells at the 72-h time point was set to 100%. Data represent the averages of all experiments ($n$ = 5 independent cultures, means ± s.e.m.). **(B, C, E, G)** Data were analyzed by a two-tailed $t$ test (C), one-way ANOVA with Dunnett's multiple comparison test (B, E), and Tukey's multiple comparison test (G). Source data are available for this figure.

Ppg1$^{H111N}$, a catalytically inactive mutant (Figs 6F and G and S6A and B), suggesting that mitophagy suppression by the OMM-anchored Far complex requires Ppg1 phosphatase activity.

### Msp1 is required for efficient mitophagy in cells lacking Get3

Previous studies demonstrate that Msp1, an OMM-anchored AAA-ATPase acting as an extractase, is important to remove non-mitochondrial TA proteins from the surface of mitochondria in the absence of Get components (Chen et al, 2014; Okreglak & Walter, 2014; Wang et al, 2020; Matsumoto et al, 2022). Accordingly, we asked whether loss of Msp1 exacerbates mitophagy deficiencies in cells lacking the GET pathway. As double knockout of Msp1 and Get1/2 elicited extremely severe growth defects under respiratory conditions, we performed fluorescence microscopy and mitophagy assays for *msp1 get3*-double-null cells that could grow slowly with relatively mild phenotypes in liquid non-fermentable medium (Fig S6C). Single knockout of Msp1 and Get3 slightly affected mitophagy (85% and 93%, respectively, compared with WT cells), whereas loss of these two proteins significantly compromised mitophagy (48% compared with WT cells) (Fig 7A and B). In addition, loss of Get3 in cells expressing Msp1$^{E193Q}$ (Msp1 EQ, an ATPase-inactive mutant) also synergistically disturbed degradation of mitochondria (51% compared with WT cells) (Fig 7A and B). These results suggest that Msp1 ATPase activity is critical to prevent mitophagy suppression in GET-deficient cells.

Next, we performed fluorescence microscopy and found that loss of Msp1 did not significantly affect ER localization of Far8-3×GFP (Fig 7C and D). In contrast, Far8-3×GFP localized to mitochondria in *get3*-null and *msp1 get3*-double-null cells (Fig 7C and D). Based on these observations, we investigated whether loss of Ppg1 affects mitophagy in *msp1 get3*-double-null cells and found that *msp1 get3 ppg1*-triple-null cells significantly restored mitophagy (81% compared with WT cells) (Fig 7E and F). Similarly, the expression of Atg32 (Δ151–200), a deletion mutant lacking a domain required for Ppg1-mediated dephosphorylation, also increased mitophagy in cells lacking Get3 and Msp1 (60% compared with WT cells) (Fig S6D and E). Furthermore, forced ER targeting of the Far complex in *msp1 get3*-double-null cells clearly increased mitophagy (90% compared with WT cells) (Fig 7G and H). Collectively, these results support the idea that the GET pathway and Msp1 cooperatively act to prevent excess accumulation of the Far complex on the OMM and oversuppression of mitophagy.

## Discussion

In the present study, we show that the GET pathway contributes to Atg32 phosphorylation by promoting localization of the Ppg1-Far phosphatase complex to the ER (Fig 8). Loss of Get1/2 (ER membrane–anchored insertase), or Get3 (cytosolic ATPase), partially reduces Atg32 phosphorylation, thereby abrogating Atg32-Atg11 interactions in the early phase of respiratory growth (Fig 1A, B, and D). Consistent with this observation, mitophagy is severely compromised in *get1/2*-null cells under prolonged respiration (Onishi et al, 2018). However, cells lacking Get3 exhibit only minor defects in mitophagy (Onishi et al, 2018), raising the possibility that in the prolonged phase of respiratory growth, Get1/2 may have unappreciated additional

function(s) to promote mitophagy independently of its insertase activity (Fig S2D–F), or that unknown protein(s) may exert a Get3-related compensatory role in promoting mitophagy.

Evidently, Atg32-Atg11 interactions and mitophagy in cells lacking Get1/2 can mostly be restored by additional loss of Ppg1, a phosphatase that dephosphorylates Atg32 (Fig 2A–F), suggesting that Ppg1 is likely to be the primary cause of reduced Atg32 phosphorylation in *get1/2*-null mutants. Consistent with these findings, loss of Far9, a component of the Far complex that binds to Ppg1 and acts in a cooperative manner to dephosphorylate Atg32, also increased Atg32-Atg11 interactions and mitophagy in Get1/2-deficient cells (Fig 4A–C). Far9 has been suggested to be an ER-resident TA protein of the Far complex (Pracheil & Liu, 2013), and loss of the Get1/2 insertase activity perturbs ER localization of the Far complex (Figs 3A–D and S2A and D), supporting the idea that the GET pathway promotes insertion of the Far TA proteins to the ER membrane.

Although disruption of the GET pathway leads to targeting of multiple ER-resident TA proteins to mitochondria (Schuldiner et al, 2008; Jonikas et al, 2009), how these ectopically targeted proteins impact events on the mitochondrial surface remains enigmatic. Upon loss of Get components, the Far complex predominantly targets to mitochondria (Figs 3A–D and S2A and D) in a manner dependent on the TA proteins Far9 and Far10 (Fig S3A and B). Anchoring to the mitochondrial surface seems to be critical for the Far complex to efficiently abrogate Atg32-Atg11 interactions and mitophagy, because cytosolic diffusion or GET-independent ER anchoring of the Far complex leads to restoration of those processes in *get1/2*-null cells (Figs 4A–C, 5C–E, and S4A–C). Based on the observations from us (Fig S3C and D) and others (Innokentev et al, 2020) that a fraction of the Far complex localizes to mitochondria even in WT cells, we favor a hypothetical model that dynamic changes in the GET pathway (e.g., expression level, insertase activity, and substrate affinity) could affect the number of Ppg1-Far complex targeted to the ER or mitochondria, thereby serving as a regulatory process for mitophagy (Fig 8). Further studies are needed to test this hypothesis.

During the course of this study, we noticed that our several results seem to be somewhat different from the recently reported data on localization and function of the Ppg1-Far complex (Innokentev et al, 2020). First, we demonstrate that the Far complex is mostly localized to the ER in cells during non-fermentable growth (Fig 3A), whereas it has been shown that the Far complex is distributed almost equally to both mitochondria and the ER in cells during fermentable growth (Innokentev et al, 2020). These distinct features might be due to different growth conditions (mitophagy-inducing or mitophagy–non-inducing). Second, cells containing the GET-independently ER-localized Far complex exhibit mitophagy at near WT levels under prolonged respiration (Fig 5D and E), whereas cells containing the Far9-Cyb5$^{TA}$-dependently ER-localized Far complex have been shown to accelerate mitophagy at the early stationary phase (Innokentev et al, 2020). These differences might result from TM segments (one derived from the non-TA protein Sec12 or TA protein Cyb5) and/or mitophagy assay time points (72 or 40 h in non-fermentable medium). Third, we show that Gem1 TA-dependent artificial targeting of the Far complex to mitochondria causes a partial defect in mitophagy under prolonged respiration (Fig 6D and E), whereas it has been demonstrated that mitochondria-targeted Far complex by Tom5 TA strongly diminishes

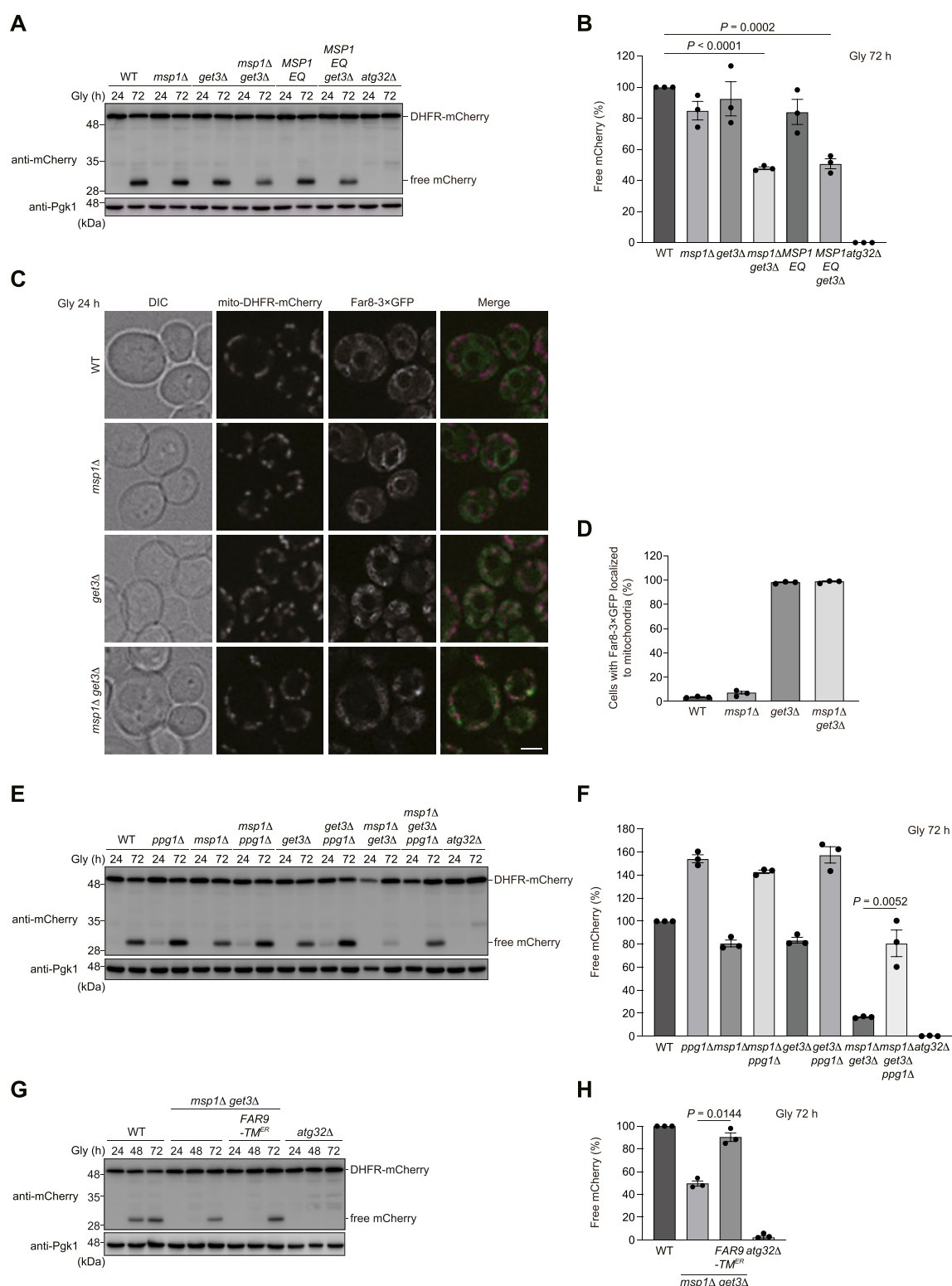

**Figure 7. Loss of Msp1 leads to synthetic mitophagy deficiencies in *get3*-null cells.**
**(A)** WT, *msp1Δ*, *get3Δ*, *msp1Δ get3Δ*, *MSP1 E193Q* (*MSP1 EQ*)-expressing, *MSP1 EQ*-expressing *get3Δ*, and *atg32Δ* cells were grown in glycerol medium (Gly), collected at the indicated time points, and subjected to Western blotting. All strains were derivatives expressing mito-DHFR-mCherry. **(A, B)** Amounts of free mCherry in cells analyzed in (A) were quantified in three experiments. The signal intensity value of free mCherry in WT cells at the 72-h time point was set to 100%. Data represent the averages of all experiments (*n* = 3 independent cultures, means ± s.e.m.). **(C)** Representative images of WT, *msp1Δ*, *get3Δ*, and *msp1Δ get3Δ* cells expressing mito-DHFR-mCherry and Far8-3×GFP grown for 24 h in glycerol medium (Gly) and observed by structured illumination microscopy. Single-plane images are shown. Scale bar, 2 µm. **(C, D)** Cells

mitophagy at the early stationary phase (Innokentev et al, 2020). This phenotypic difference might be attributed to TA domains used for mitochondrial anchoring and/or mitophagy assay time points (72 or 40 h in non-fermentable medium). Nevertheless, it seems possible that the mitochondria-anchored Ppg1-Far complex could suppress stationary-phase mitophagy more effectively at the early phase than at the late phase.

The expression of the Get1/2 insertase–inactive mutants leads to extensive accumulation of the Ppg1-Far complex on the mitochondrial surface, whereas mitophagy is only partially decreased in these mutant cells (70% compared with WT cells) (Fig S2D–F). Notably, these phenotypes are similar to those in *get3*-null cells (Fig 7A–D) (Onishi et al, 2018), which is in agreement with the previous finding that the Get1/2 insertase–inactive mutants cannot recruit Get3 to the ER (Wang et al, 2011). In addition, artificial targeting of the Ppg1-Far complex to mitochondria only partially reduces mitophagy under prolonged respiration (70% compared with WT cells) (Fig 6D and E). Together, these findings raise the possibility that Get1/2 may be a bifunctional complex acting as a general insertase for ER-resident TA proteins and serving as a pro-mitophagic factor independently of its insertase activity.

Finally, our data reveal a potential role of the OMM-anchored AAA-ATPase Msp1 in mitophagy. Consistent with the previous reports that Msp1 extracts non-mitochondrial TA proteins from the mitochondrial surface upon loss of Get components (Chen et al, 2014; Okreglak & Walter, 2014; Wohlever et al, 2017; Wang et al, 2020; Matsumoto et al, 2022), cells lacking both Get3 and Msp1 display synthetic defects in mitophagy that can be rescued by loss of Ppg1, expression of an Atg32 variant defective in its interaction with the Ppg1-Far complex, or GET-independent ER anchoring of the Far complex (Figs 7A, B, and E–H and S6D and E). Thus, although ER localization of the Far complex seems to be hardly altered in cells lacking Msp1 (Fig 7C and D), it remains possible that this OMM-anchored extractase may act in removal of ER/mitochondrial TA proteins, such as Far9 and Far10, thereby contributing to Atg32 phosphorylation, Atg32-Atg11 interactions, and mitophagy (Fig 8). How the GET pathway and Msp1 coordinately act in activation of Atg32-mediated mitophagy awaits further investigations.

# Materials and Methods

## Yeast strains and plasmids used in this study

Yeast strains and plasmids used in this thesis are listed in Tables S1 and S2. Standard genetic and molecular biology methods were performed for generating yeast strains.

## Growth conditions of yeast

Yeast cells were incubated in YPD medium (1% yeast extract, 2% peptone, and 2% dextrose), and synthetic medium (0.17% yeast nitrogen base without amino acids and ammonium sulfate, 0.5% ammonium sulfate) with 0.5% casamino acids and either 2% dextrose (SDCA) or 0.1% dextrose plus 3% glycerol (SDGCA), supplemented with the necessary amino acids. For mitophagy assay under respiratory conditions, cells grown to mid-log phase in SDCA were transferred to SDGCA and incubated at 30°C.

## Protein phosphatase treatment assays

For protein phosphatase assays, cells were pregrown in SDCA and transferred to SDGCA. 2.0 $OD_{600}$ units of cells were collected and subjected to alkaline lysis and TCA (trichloroacetic acid) precipitation. The pellet was resuspended in a reaction buffer (50 mM Tris–HCl, pH 7.5, 100 mM NaCl, 2 mM DTT, 0.5 mM EDTA, 0.01% Brij-35, and 2 mM $MgCl_2$), treated with or without λ protein phosphatase (λ-PPase) in the presence or absence of PPase inhibitor at 30°C for 1 h. Samples corresponding to 0.2 $OD_{600}$ units of cells were loaded per lane.

## Structured illumination microscopy

Live yeast cells expressing Far8-3×GFP were observed using structured illumination microscopy. Differential interference contrast and fluorescence images were obtained under a KEYENCE BZ-X810 system equipped with a 100× objective lens (CFI Apochromat TIRF 100XC Oil, Plan-APO TIRF 100, NA: 1.49; Nikon), filter sets for GFP and mCherry (BZ-X filter GFP and BZ-X filter TRITC, respectively; KEYENCE). Cell images were captured using acquisition and analysis software (BZ-X800 Analyzer; KEYENCE).

## Western blotting

Samples corresponding to 0.1–0.4 $OD_{600}$ units of cells were separated by SDS–PAGE followed by Western blotting and immunodecoration with primary antibodies raised against mCherry (1:2,000, ab125096; Abcam), Pgk1 (1:10,000, ab113687; Abcam), GFP (1:1,000, 13921700; Roche), HA (1:5,000, A2095; Sigma-Aldrich), and Atg11 (1:1,000, gift from Dr. Hayashi Yamamoto, Nippon Medical School). After treatment with the secondary antibodies, HRP-conjugated rabbit anti-mouse IgG (H + L) for mCherry, GFP, HA, and Pgk1, and goat anti-rabbit mouse IgG (H + L) for Atg11 (1:10,000, 315-035-003 and 111-035-003, respectively; Jackson ImmunoResearch) followed by the enhanced chemiluminescence reagent Western Lightning Plus-ECL (203-19151; PerkinElmer) or

---

analyzed in (C) were quantified in three experiments. Data represent the averages of all experiments (*n* = 3 independent cultures, means ± s.e.m.). **(E)** WT, *ppg1Δ*, *msp1Δ*, *get3Δ*, *msp1Δ ppg1Δ*, *get3Δ ppg1Δ*, *msp1Δ get3Δ*, *msp1Δ get3Δ ppg1Δ*, and *atg32Δ* cells expressing mito-DHFR-mCherry were grown in glycerol medium (Gly), collected at the indicated time points, and subjected to Western blotting. **(E, F)** Amounts of free mCherry in cells analyzed in (E) were quantified in three experiments. The signal intensity value of free mCherry in WT cells at the 72-h time point was set to 100%. Data represent the averages of all experiments (*n* = 3 independent cultures, means ± s.e.m.). **(G)** WT, *msp1Δ get3Δ*, and *atg32Δ* cells expressing mito-DHFR-mCherry and WT *FAR9* or *FAR9-TM^ER^* were grown in glycerol medium (Gly), collected at the indicated time points, and subjected to Western blotting. **(G, H)** Amounts of free mCherry in cells under respiratory conditions for 72 h in (G) were quantified in three experiments. The signal intensity value of free mCherry in WT cells at the 72-h time point was set to 100%. Data represent the averages of all experiments (*n* = 3 independent cultures, means ± s.e.m.). **(B, F, H)** Data were analyzed by one-way ANOVA with Dunnett's multiple comparison test (B) or a two-tailed *t* test (F, H).
Source data are available for this figure.

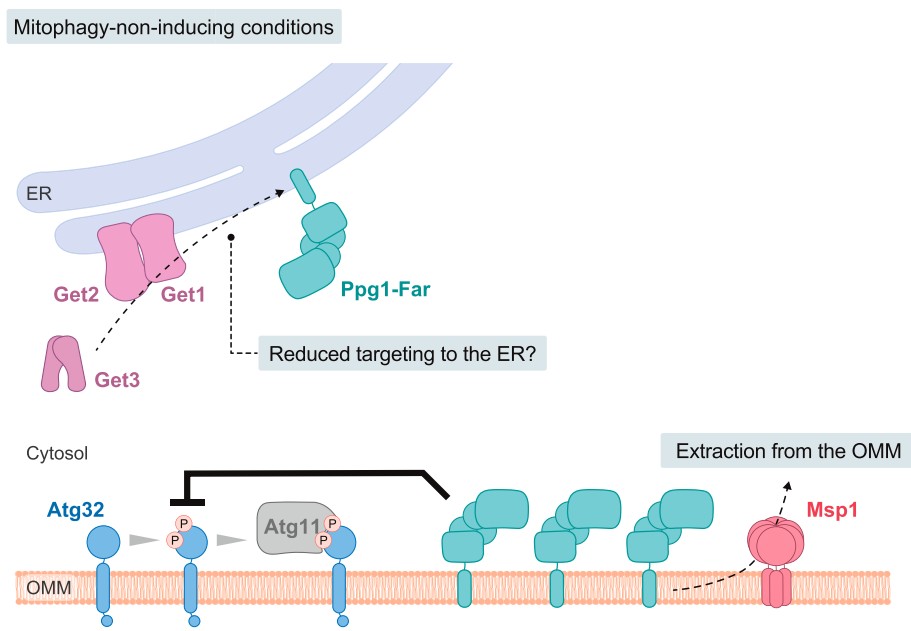

Mitophagy-non-inducing conditions

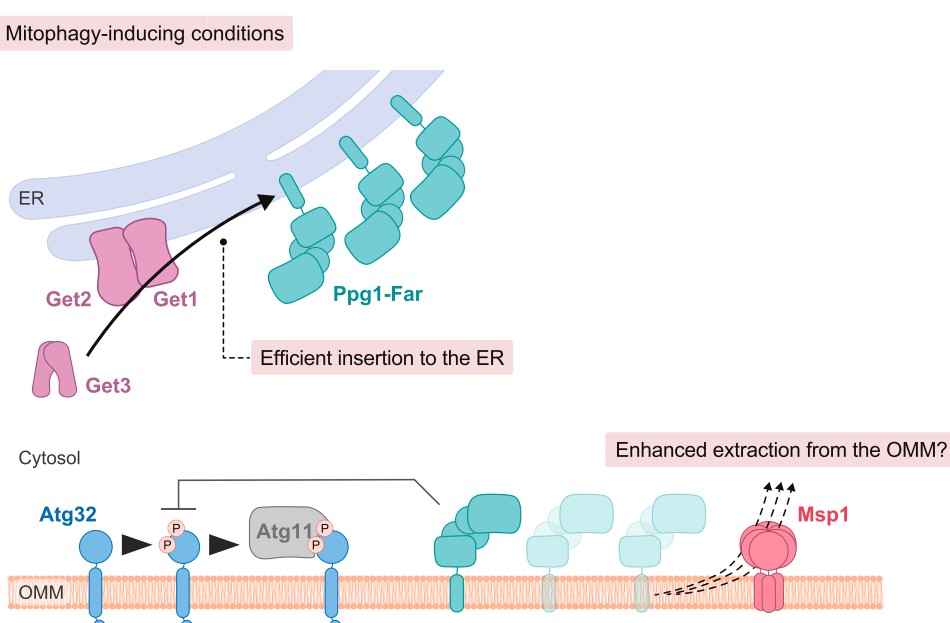

Mitophagy-inducing conditions

**Figure 8. Hypothetical model for activation of Atg32-mediated mitophagy.**
(Upper panel) Under mitophagy-non-inducing (fermentable) conditions, a substantial fraction of the Ppg1-Far complex escapes the GET pathway, localizes to mitochondria, and suppresses Atg32 phosphorylation and Atg32-Atg11 interactions. Mitochondria-anchored Ppg1-Far can be extracted from the OMM via Msp1. (Lower panel) Under mitophagy-inducing (non-fermentable) conditions, the GET pathway efficiently mediates targeting of the Ppg1-Far complex to the ER, which in turn promotes Atg32 phosphorylation and Atg32-Atg11 interactions. Msp1-dependent extraction of mitochondria-anchored Ppg1-Far from the OMM can contribute to activation of mitophagy.

ImmunoStar LD (PTJ2005; Wako), proteins were detected using a luminescent image analyzer (FUSION Solo S; VILBER). Quantification of the signals was performed using FUSION Solo S (VILBER).

### Bioluminescence assay for protein–protein interactions

For quantitative analysis of Atg32-Atg11 interactions using NanoBiT (Promega), Atg32 fused to three copies of GFP and Large BiT (LgBiT; 17.6 kD), and Atg11 fused to Small BiT (SmBiT; 11 amino acids) were expressed endogenously (constructed by Yang Liu, Osaka University). Upon interaction of Atg32 with Atg11, SmBiT and LgBiT are brought into close proximity, leading to structural complementation and generation of a luminescent signal. For the assay, cells were grown in glycerol media (SDGCA). 1.0 $OD_{600}$ units of cells were collected in the early phase of respiration ($OD_{600}$: 1.4–1.6) and washed with 400 µl PBS. After washing, cells were dissolved in 40 µl PBS and applied to a 96-well plate. The detection reagent was prepared by diluting the Nano-Glo Live Cell Substrate (0000360026;

Promega) with the Nano-Glo LCS Dilution Buffer (0000333050; Promega) to make the Nano-Glo Live Cell Reagent. 10 μl diluted detection reagent was added onto the 96-well plate and mixed with the cells. Then, cells were incubated at 30°C for 1 h. After incubation, the luminescent signal was detected by the microplate reader (Fluoroskan Ascent FL; Thermo Fisher Scientific) using a filter (exposure time: 1,000 ms). For the detection of the GFP fluorescent signal derived from Atg32-3HA-3×GFP-3FLAG-LgBiT, 1.0 $OD_{600}$ units of cells were collected at the same time point, and dissolved in 100 μl SDGCA media, applied to a 96-well plate. GFP signal was measured by a microplate reader (Fluoroskan Ascent FL; Thermo Fisher Scientific) (excitation: 485 nm, emission: 538 nm, exposure time: 1,000 ms). The resultant luminescent intensity was normalized by the GFP signal.

## Statistical analysis

Results are presented as means including means ± s.e.m. Statistical analyses were performed with Excel for Mac (Microsoft) and GraphPad Prism 9 (GraphPad Software), using a two-tailed $t$ test and one-way ANOVA followed by Tukey's or Dunnett's multiple comparison test. All the statistical tests performed are indicated in the figure legends.

## Supplementary Information

## Acknowledgements

We thank Miyuki Sato (Gunma University, Japan) for valuable suggestions on artificial ER anchoring, Elmar Schiebel (Heidelberg University, Germany) for kindly providing us with the plasmid pFA6a-3myeGFP-kanMX6, and Yang Liu (Osaka University, Japan) for providing us with the NanoBiT assay strains. We are also grateful to Hayashi Yamamoto (Nippon Medical School, Japan) for kind gifts of antibodies. This work was supported in part by JSPS KAKENHI Grants JP19J10384 and JP21K15041 (to M Onishi), and JP16H04784, JP19H03222, and JP20H05324 (to K Okamoto); and the Osaka University International Joint Research Promotion Programs (Type A+ and Type A-GKP) (to K Okamoto).

### Author Contributions

M Onishi: conceptualization, data curation, formal analysis, funding acquisition, investigation, visualization, methodology, and writing—original draft.
M Kubota: data curation, formal analysis, investigation, and visualization.
L Duan: data curation, formal analysis, investigation, and visualization.
Y Tian: data curation, formal analysis, investigation, and visualization.
K Okamoto: conceptualization, data curation, supervision, funding acquisition, methodology, project administration, and writing—original draft, review, and editing.

### Conflict of Interest Statement

The authors declare that they have no conflict of interest.

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
