## [Reviewer comments · Life Science Alliance]

Life Science Alliance

The GET pathway serves to activate Atg32-mediated mitophagy by ER targeting of the Ppg1-Far complex

Mashun Onishi, Mitsutaka Kubota, Lan Duan, Yuan Tian, and Koji Okamoto

DOI: <https://doi.org/10.26508/lsa.202201640>

Corresponding author(s): Koji Okamoto, Osaka University

Review Timeline:

Submission Date:	2022-07-31
Editorial Decision:	2022-09-01
Revision Received:	2022-11-28
Editorial Decision:	2023-01-03
Revision Received:	2023-01-09
Accepted:	2023-01-09

Scientific Editor: Novella Guidi

Transaction Report:

September 1, 2022

Re: Life Science Alliance manuscript #LSA-2022-01640

Prof. Koji Okamoto
Osaka University
Graduate School of Frontier Biosciences
1-3 Yamadaoka
Suita, Osaka 565-0871
Japan

Dear Dr. Okamoto,

Thank you for submitting your manuscript entitled "The GET pathway serves to activate Atg32-mediated mitophagy by ER targeting of the Ppg1-Far complex" to Life Science Alliance. The manuscript was assessed by expert reviewers, whose comments are appended to this letter. We invite you to submit a revised manuscript addressing the Reviewer comments.

Thank you for this interesting contribution to Life Science Alliance. We are looking forward to receiving your revised manuscript.

Sincerely,

B. MANUSCRIPT ORGANIZATION AND FORMATTING:

Reviewer #1 (Comments to the Authors (Required)):

This manuscript describes how ATG32-ATG11 driven mitophagy is attenuated due to decreased ATG32 phosphorylation caused by loss of the GET genes that causes the Ppg1-Far phosphatase to mis-localize to mitochondria and de-phosphorylate ATG32 aberrantly, causing defective mitophagy. Deletion of Ppg1 or the Far proteins rescued the ATG32-ATG11 interaction and defective mitophagy caused by GET deletion. Mitochondrial localization of the Ppg1-Far complex relied on the TA domains of Far complex proteins. Conversely, artificial tagging of Far proteins to the ER promoted mitophagy while their artificial targeting to the mitochondria inhibited mitophagy. The ability of mitochondrial Ppg1-Far complex to suppress mitophagy was dependent on the phosphatase activity of Ppg1. Finally they show that the Msp1 AAA+ ATPase can partially compensate for loss of the GET proteins by retrotranslocating Ppg1-Far complex out of the mitochondrial OMM.

Overall, this is a very careful and thoughtfully written manuscript and the western data and imaging data shown are convincing up to a point.

There key concern that should be addressed is that the work relies extensively on the NanoBiT assay for all protein interaction work and also that most of the work involves use of over-expression mutants and reporter assays, and lacks examination of endogenous proteins. It seems that the work could benefit from quantification of endogenous markers of mitophagy, for example even measuring changes in mitochondrial mass and localization of endogenous Ppg1 etc. Some grammatical matters also need addressed.

Reviewer #2 (Comments to the Authors (Required)):

Previous work by the submitting group has shown that GET pathway mutants are defective in mitophagy, but not or less in other autophagy pathways, suggesting a specialized function for the GET pathway in mitophagy (Onishi BBR 2018). Work by others has previously shown that CK2 mediated phosphorylation of Atg32 promotes Atg11 binding and mitophagy, whereas Ppg1-Far counteracts this by dephosphorylating Atg32 (Furukawa Cell Rep 2018). The same group then further showed that Ppg1-Far complex proteins Far9 and Far10 are tail-anchored proteins, which localize Ppg1-Far to the ER and mitochondria. A forced ER localization of the Far complex results in no inhibition of mitophagy, whereas a forced mitochondrial localization does, suggesting that the Far complex plays different roles, at the ER in TORC2 signaling, and at the mitochondria in mitophagy, by dephosphorylating Atg32 and thereby inhibiting mitophagy (Innokentev eLife 2020). It has also been shown that the GET pathway regulates ER insertion of tail-anchored proteins by various studies, and defects in this pathway result in tail-anchored proteins localizing to mitochondria, where Mps1 can remove these. Together, these published observations propose that the GET pathway regulates mitophagy by regulating Far protein insertion into the ER and mitochondria, which regulates Ppg1-Far phosphatase action on Atg32 and therefore Atg32-Atg11 binding. This is the question addressed in this manuscript.

In the presented work the authors nicely and convincingly show that the GET pathway is required to localize Ppg1-Far to the ER, and if it fails to do so, Ppg1-Far is enhanced at mitochondria, resulting in Atg32 dephosphorylation and mitophagy inhibition. Msp1, however, can counteract this increased mitochondrial localization of the Ppg1-Far complex and thereby restore mitophagy function. Together, these findings suggest that the GET pathway and Msp1 cooperate to regulate mitophagy by controlling the localization of the Ppg1-Far complex.

This is a nice mechanistic study using classical knockout approaches as well as anchor-away strategies to restore/prevent function. I believe this work is well suited for Life Science Advances and provides valuable new information for the field. The experiments shown are very well done and only little revision is needed. I have only two points that should be addressed, and one further suggestion:

1. Several experiments use the expression of Atg32 mutants and tagged versions of Atg11. However, expression levels are mostly not checked, neither for Atg32 nor for Atg11. If a mutant is unstable or Atg11 in the presence of a mutant is unstable, this would result in a similar situation as a knock-out, and could lead to similar results. Therefore, experiments should be controlled for expression levels in some way. Ideal would be antibodies detecting these proteins, if not available the authors could at least check if a tagged version of these mutants expresses to similar amounts as the wild type.

2. Similar to the first point, also for GET mutants no expression levels are shown. If the authors want to claim that the effect observed is due to insertase activity, they need to show that the inactive mutant actually is expressed to similar levels as the wild type protein. If not possible to show due to for instance lack of antibodies, then this statement should be removed.

3. The data presented proposes a cooperation between the GET pathway and Mps1. What has not been addressed is which of the two is the major regulator for induction of mitophagy under non-mutant conditions when cells switch from 'rich' conditions to mitophagy conditions. With the assays and tools the authors have established, they could clarify this point.

Thank you very much for handling our manuscript and inviting us to submit a revised version. First of all, we deeply apologize for our delayed submission. We have been trying to catch up multiple things that have been accumulated in our to-do list under the COVID-19 situations, and are now finally ready to submit a revised manuscript. We have carefully read all remarks made by the Reviewers. We truly appreciate the critical but constructive comments, and have addressed the issues in the revised manuscript and this accompanying letter. Please see the following point-by-point responses to the comments below.

First, we added a summary blurb in the first page of our revised manuscript.

(page 1, line 19): **The GET pathway promotes anchoring of the Ppg1-Far phosphatase complex to the ER membrane, contributing to initiation of Atg32-mediated mitochondria-specific autophagy in budding yeast.**

Reviewer #1's comments

We appreciate all the points and have carefully addressed them as described below.

There key concern that should be addressed is that the work relies extensively on the NanoBiT assay for all protein interaction work and also that most of the work involves use of over-expression mutants and reporter assays, and lacks examination of endogenous proteins. It seems that the work could benefit from quantification of endogenous markers of mitophagy, for example even measuring changes in mitochondrial mass and localization of endogenous Ppg1 etc.

We thank the reviewer for raising these points up. In this study, we sought to perform a NanoBiT assay for evaluating Atg32-Atg11 interactions in Get-mutant cells, instead of using conventional methods such as immunoprecipitation (IP). The NanoBiT system enables us to quantitatively analyze protein-protein interactions based on the luminescent signals in live cells, which cannot be done with IP.

For mitophagy assays, we used strains constitutively expressing a mitochondrial matrix-targeted DHFR (dihydrofolate reductase)-mCherry (mito-DHFR-mCherry) reporter protein. In this study, we did not examine endogenous mitochondrial markers and quantify them to evaluate mitophagy. However, in our previous study (Onishi et al., *BBRC*, 503: 14-20 [2018]), we quantified the amount of Por1, an outer mitochondrial membrane protein, as an endogenous marker to monitor changes in mitochondrial mass. Under mitophagy-inducing conditions, we observed a robust decrease in the Por1 protein levels in Get-mutants, which nicely correlates with the vacuolar protease-dependent processing of mito-DHFR-mCherry that generates free mCherry. Thus, the amount of free mCherry reflects a reduction in mitochondrial mass via mitophagy, and we sought to quantify mitophagy using mito-DHFR-mCherry processing assays.

About analyzing Ppg1 localization, we found that GFP-tagged Ppg1 is not fully functional, and therefore monitored functionally 3×GFP-tagged Far8 (a component of the Ppg1-Far complex) to indirectly visualize Ppg1 using fluorescence microscopy.

Reviewer #2's comments

We appreciate all the points and have carefully addressed them with additional sentences in the main text and modified figure as described below.

(1) Several experiments use the expression of Atg32 mutants and tagged versions of Atg11. However, expression levels are mostly not checked, neither for Atg32 nor for Atg11. If a mutant is unstable or Atg11 in the presence of a mutant is unstable, this would result in a similar situation as a knock-out, and could lead to similar results. Therefore, experiments should be controlled for expression levels in some way. Ideal would be antibodies detecting these proteins, if not available the authors could at least check if a tagged version of these mutants expresses to similar amounts as the wild type.

We checked the expression levels of LgBiT-tagged Atg32 and SmBiT-tagged Atg11, and confirmed that those protein levels were not significantly reduced upon loss of Get proteins (Fig. S1B and C). In addition, we demonstrated that expression profiles of an Atg32 mutant lacking 151-200 amino acid residues were not significantly altered in wild-type and *get1/2*-null mutant cells (Fig. S1G). Accordingly, we modified the sentences in the main text and figure legends as described below.

(page 6, line 127): **Reduction in Atg32-Atg11 interactions did not seem to be mainly caused by a decrease in Atg32 and Atg11 expression levels (Fig. S1B and C).**

(page 7, line 161): **We also confirmed that expression of the Atg32 mutant (Δ 151-200) does not significantly change in *get1/2*-null cells (Fig. S1G), suggesting that these phenotypes are not mainly caused by aberrant Atg32 levels.**

(page 33, line 768): **(B) Wild-type, *get1* Δ , *get2* Δ , and *get3* Δ cells expressing Atg32-3HA-3xGFP-3FLAG-LgBiT and Atg11-HA-SmBiT, and the original BY4741 negative control (N.C.) pregrown in fermentable dextrose medium (Dex) were cultured in non-fermentable glycerol medium (Gly), collected at the indicated OD₆₀₀ points, and subjected to western blotting. (C) Wild-type, *get1* Δ , *get2* Δ , and *get3* Δ cells expressing Atg32-3HA-3xGFP-3FLAG-LgBiT and Atg11-HA-SmBiT, and *atg11* Δ cells pregrown in fermentable dextrose medium (Dex) were cultured in non-fermentable glycerol medium (Gly), collected at the indicated OD₆₀₀ points, and subjected to western blotting.**

(page 33, line 787): **(G) Wild-type, *get1* Δ , and *get2* Δ cells expressing chromosomally integrated *ATG32-3HA* wild-type or *ATG32 (Δ 151-200)-3HA* (indicated as *Δ 151-200-3HA*), and *atg32* Δ cells were grown in glycerol medium (Gly), collected at the indicated OD₆₀₀ points, and subjected to western blotting. All strains are *atg7*-null derivatives defective in degradation of Atg32-3HA via mitophagy.**

(2) Similar to the first point, also for GET mutants no expression levels are shown. If the authors want to claim that the effect observed is due to insertase activity, they need to show that the inactive mutant actually is expressed to similar levels as the wild type protein. If not possible to show due to for instance lack of antibodies, then this statement should be removed.

We checked the expression levels of Get1/2 insertase-inactive mutants and confirmed that expression levels of these protein levels were almost comparable to wild-type Get1 or Get2 (Fig. S2B and C). We also added the sentences in the main text and figure legends as described below.

(page 8, line 185): **To clarify whether the insertase activity of Get1/2 is required for Far8-3xGFP localization to the ER, we generated yeast strains expressing an inactive Get1 or Get2 variant with point mutations in their conserved cytosolic domain (Get1NRm: N72A, R73A, Get2RERRm: R14E, E15R, R16E, R17E) (Wang et al, 2011), and confirmed that these mutants are expressed at near wild-type levels (Fig. S2B and C). Using these strains, we found that expression of these insertase-inactive mutants significantly disturbed ER localization of Far8-3xGFP (95% and 98% of cells expressing Get1NRm and Get2RERRm, respectively) (Fig. S2D), further underscoring a primary role for the GET pathway in ER targeting of the Ppg1-Far complex.**

(page 34, line 797): **(B) Wild-type cells expressing GFP-tagged *GET1* or *GET1 NRm (N72A, R73A)* were grown in glycerol medium (Gly), collected at the indicated time points, and subjected to western blotting. (C) Wild-type cells expressing 3×GFP-tagged *GET2* or *GET2 RERRm (R14E, E15R, R16E, R17E)* were grown in glycerol medium (Gly), collected at the indicated time points, and subjected to western blotting.**

(3) The data presented proposes a cooperation between the GET pathway and Msp1. What has not been addressed is which of the two is the major regulator for induction of mitophagy under non-mutant conditions when cells switch from 'rich' conditions to mitophagy conditions. With the assays and tools the authors have established, they could clarify this point.

This is an interesting point to elucidate how the GET pathway and Msp1 are coordinated to induce mitophagy.

To address this point, we first examined if the Get1/2 insertase activity is changed under respiratory conditions. We found that Get2, but not Get1, is phosphorylated during prolonged respiration, and further identified the amino acid residue important for this post-translational modification (in the attached Figure 1-A and B). We expected that this modification may affect the Get1/2 insertase activity and mitophagy induction. However, mitophagy in cells expressing a non-phosphorylated Get2 mutant is almost comparable to wild-type cells (in the attached Figure 1-C).

In addition, we attempted to detect an interaction between Msp1 and Far8 and examined whether these interactions are changed upon mitophagy. However, our immunoprecipitation assay could not detect Far8 interacting with Msp1 wild-type or Msp1(E193Q) mutant (in the attached Figure 2-A). In addition, we sought to detect Msp1-Far8 interactions in Get3-deficient cells, where Far8 is accumulated on the mitochondrial membranes. However, we could not find Far8 co-precipitated with Msp1 even in Get3-deficient cells (in the attached Figure 2-B), making it difficult to assess the interactions between Far8 and Msp1, and examine if these interactions are regulated upon mitophagy induction.

We, once again, thank the reviewers for many valuable comments. We think that our revised manuscript with the extensive modifications has significantly been improved, and hope that the paper will be acceptable for publication in *Life Science Alliance*.

1. Significance of Get2 phosphorylation on mitophagy

1-A. Get2, but not Get1, is phosphorylated upon respiration

1-B. Identification of amino acid residues important for Get2 phosphorylation

1-C. Mitophagy in cells expressing a Get2 S60A mutant

Mitophagy assay

2. Msp1-Far8 interactions

2-A. Immunoprecipitation to detect interactions between Msp1 and Far8

Immunoprecipitation

Onishi et al.

Figure for revision (LSA-2022-01640) related to Reviewer #2's comments

2. Msp1-Far8 interactions *continued*

2-B. Immunoprecipitation to detect interactions between Msp1 and Far8 in cells lacking Get3

Immunoprecipitation

January 3, 2023

RE: Life Science Alliance Manuscript #LSA-2022-01640R

Prof. Koji Okamoto
Osaka University
Graduate School of Frontier Biosciences
1-3 Yamadaoka
Suita, Osaka 565-0871
Japan

Dear Dr. Okamoto,

Thank you for submitting your revised manuscript entitled "The GET pathway serves to activate Atg32-mediated mitophagy by ER targeting of the Ppg1-Far complex". We would be happy to publish your paper in Life Science Alliance pending final revisions necessary to meet our formatting guidelines.

- please upload your main manuscript text as an editable doc file
- please upload your main and supplementary figures as single files
- please add the Twitter handle of your host institute/organization as well as your own or/and one of the authors in our system
- please use the [10 author names, et al.] format in your references (i.e. limit the author names to the first 10)
- the Online Supplemental Material section is unnecessary and should be removed

A. FINAL FILES:

B. MANUSCRIPT ORGANIZATION AND FORMATTING:

**Submission of a paper that does not conform to Life Science Alliance guidelines will delay the acceptance of your

manuscript.**

The license to publish form must be signed before your manuscript can be sent to production. A link to the electronic license to publish form will be sent to the corresponding author only. Please take a moment to check your funder requirements.

Sincerely,

Reviewer #2 (Comments to the Authors (Required)):

The authors have addressed and clarified all points raised. I have no further comments and support publication.

January 9, 2023

RE: Life Science Alliance Manuscript #LSA-2022-01640RR

Prof. Koji Okamoto
Osaka University
Graduate School of Frontier Biosciences
1-3 Yamadaoka
Suita, Osaka 565-0871
Japan

Dear Dr. Okamoto,

Thank you for submitting your Research Article entitled "The GET pathway serves to activate Atg32-mediated mitophagy by ER targeting of the Ppg1-Far complex". It is a pleasure to let you know that your manuscript is now accepted for publication in Life Science Alliance. Congratulations on this interesting work.

DISTRIBUTION OF MATERIALS:

Again, congratulations on a very nice paper. I hope you found the review process to be constructive and are pleased with how the manuscript was handled editorially. We look forward to future exciting submissions from your lab.

Sincerely,
